# Microneedle-array patches loaded with dual mineralized protein/peptide particles for type 2 diabetes therapy

Wei Chen[1,2], Rui Tian[2,3], Can Xu[2], Bryant C. Yung[2], Guohao Wang[3], Yijing Liu[2], Qianqian Ni[2], Fuwu Zhang[2], Zijian Zhou[2,3], Jingjing Wang[2], Gang Niu[2], Ying Ma[2], Liwu Fu[1] & Xiaoyuan Chen [2]

The delivery of therapeutic peptides for diabetes therapy is compromised by short half-lives of drugs with the consequent need for multiple daily injections that reduce patient compliance and increase treatment cost. In this study, we demonstrate a smart exendin-4 (Ex4) delivery device based on microneedle (MN)-array patches integrated with dual mineralized particles separately containing Ex4 and glucose oxidase (GOx). The dual mineralized particle-based system can specifically release Ex4 while immobilizing GOx as a result of the differential response to the microenvironment induced by biological stimuli. In this manner, the system enables glucose-responsive and closed-loop release to significantly improve Ex4 therapeutic performance. Moreover, integration of mineralized particles can enhance the mechanical strength of alginate-based MN by crosslinking to facilitate skin penetration, thus supporting painless and non-invasive transdermal administration. We believe this smart glucose-responsive Ex4 delivery holds great promise for type 2 diabetes therapy by providing safe, long-term, and on-demand Ex4 therapy.

[1] State Key Laboratory of Oncology in South China, Collaborative Innovation Center for Cancer Medicine, Sun Yat-sen University Cancer Center, Guangzhou 510060, China. [2] Laboratory of Molecular Imaging and Nanomedicine (LOMIN), National Institute of Biomedical Imaging and Bioengineering (NIBIB), National Institutes of Health (NIH), Bethesda, MD 20892, USA. [3] State Key Laboratory of Molecular Vaccinology and Molecular Diagnostics, Center for Molecular Imaging and Translational Medicine, School of Public Health, Xiamen University, Xiamen 361102, China. Correspondence and requests for materials should be addressed to L.F. (email: Fulw@mail.sysu.edu.cn) or to X.C. (email: shawn.chen@nih.gov)

Diabetes mellitus is one of the most challenging health problems of the twentieth century, affecting more than 285 million people in 2010, with an expected rise to 439 million by 2030[1, 2]. A typical clinical pathological feature of diabetes is a disorder of glucose regulation, which results in severe complications such as retinopathy, nephropathy, neuropathy, and so on, significantly reducing the patient's quality of life[3]. Importantly, over 90% of diabetic individuals suffer from type 2 diabetes (T2D)[4]. Despite the long list of treatment options among an increasing number of potential drugs, more than 50% of T2D cases are not appropriately managed[5]. The failure to reach a stable glycemic status is mainly due to poor patient's compliance to prescribed treatment regimens, which often require high-frequent and intricate mealtime-related administration[6]. Moreover, most of the current prescription medications are challenged by unexpected side effects including weight gain and a risk of hypoglycemia[7].

A potential strategy to combat T2D is to utilize glucagon-like peptide-1 (GLP-1) receptor agonists, which have been recently proved to exhibit large potentials to address most therapeutic challenges[8]. The parent peptide, GLP-1, is produced and secreted by the intestinal L cells in response to nutrient ingestion[9]. It exerts functions mainly by binding and activating the pancreatic GLP-1 receptor, and thereby potentiates glucose-induced insulin secretion of β-cells[10]. However, native GLP-1 is always easily degraded by dipetidylpeptidase IV (DPP IV) to produce some N-terminally truncated metabolites in vivo, followed by rapid clearance (half-life is around 2 min), preventing its broad applications[8]. Exendin-4 (Ex4), which shares ~53% sequence homology with mammalian GLP-1, acts as a more potential GLP-1 receptor agonist by showing great affinity to the GLP-1 receptor

and high tolerance to DPP IV degradation[8, 11]. It has been approved by the U.S. Food and Drug Administration (FDA) to serve as a potent candidate for clinical T2D therapy[12, 13]. However, its blood circulation time is still relatively short (60–90 min), requiring twice-daily subcutaneous injections[14]. The high frequency of injections decrease patient's compliance, leading to limited therapeutic effect and high treatment cost[6]. Moreover, overdosing of Ex4 may result in certain adverse reactions such as dose-limiting nausea and vomiting, which diminish the quality of life[15]. All of these impediments have stimulated interest in developing an alternative treatment, which can directly respond to blood glucose level, and provide an on-demand and long-term Ex4 administration.

Currently, researchers have developed some "artificial pancreases" to provide on-demand and automated drug treatments for diabetes patients[16, 17]. Among them, the closed-loop electronic/mechanical devices that use a continuous glucose monitoring sensor and an external insulin infusion pump are promising[18]. However, such devices are mainly plagued by a high cost of the instruments, complicated device management (requirements of periodic calibration against a standard measurement), patient's discomfort (wear relatively heavy electronic devices), and high susceptibility to biofouling (need repeated drug loading and cleaning processes)[19, 20]. A chemical approach may avoid the limitation by introducing economic bio-sourced materials and acting in a disposable manner[21–23]. In chemistry, closed-loop systems for antidiabetic drug release mainly focus on stimuli-responsive materials. They can respond to the glucose-induced environmental alteration with help of glucose-sensing elements to undergo structural rearrangement (i.e., dissociation, swelling, shrinking), leading to controllable and tunable drug release[21–23].

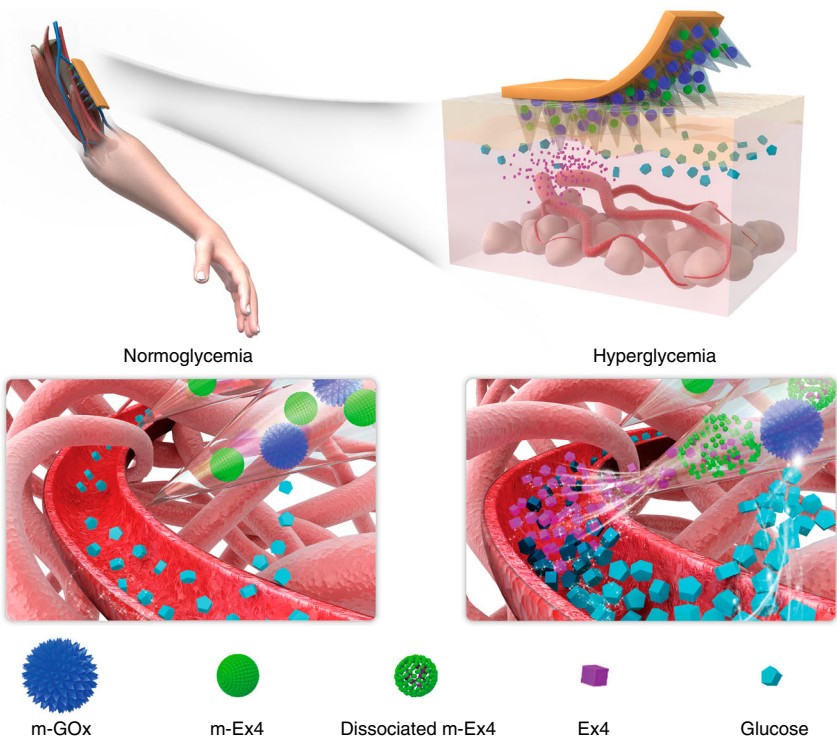

Normoglycemia          Hyperglycemia

m-GOx     m-Ex4     Dissociated m-Ex4     Ex4     Glucose

**Fig. 1** Schematic of glucose-responsive Ex4 delivery MN-array patches. After protein/peptide mineralization and MN fabrication, the patch can be directly administered onto the skin. In the normoglycemic state, relative lower glucose concentration only induces a mild pH decrease under m-GOx catalysis, which is not sufficient to induce m-Ex4 dissociation to release the payloads. In contrast, when blood glucose concentration rises to the hyperglycemic level, the significant pH decrease triggers rapid m-Ex4 dissociation to release Ex4 for blood glucose regulation, suggesting a pathology-specific drug release. Moreover, m-GOx is stable and robust in the whole process, providing long-term responsiveness to glucose change. In this manner, a smart, long-acting, and on-demand Ex4 release is achieved

For instance, phenylboronic acid (PBA) and its derivatives can effectively and reversibly interact with polyol molecules (e.g., glucose) to introduce negative charges or hydrophilicity into a polymeric network, resulting in matrix swelling, or disaggregation[24, 25]. Glucose-binding protein (GBP) (e.g., Concanavalin A) is frequently used to regulate antidiabetic drug release by providing multiple active sites for competitive binding of glucose and glycosylated moieties on the polymer chain, conferring low viscosity within the gel network[26, 27]. Alternatively, glucose oxidase (GOx) is frequently applied to catalyze the oxidation reaction of glucose to induce an acidic, hypoxic, and $H_2O_2$-rich microenvironment, which leads to degradation or dissociation of polymeric matrices[28–34]. However, in most cases, the glucose-sensing element and the drug-releasing module are integrated into the same formulation[21–23]. Through this method, the glucose-sensing element will easily leak out accompanying the dissociation of the drug-releasing module. Therefore, the formulations are inherently limited to the multi-round responsiveness of glucose challenge[35]. Moreover, protein activity loss (e.g., Concanavalin A, GOx) during storage and administration is another issue[36, 37], which significantly retards the long-term responsive capacity of the system. To develop a long-acting and closed-loop system, the strategies that can selectively stabilize and immobilize the glucose-sensing element, and separately dissociate a drug-releasing module are needed.

Herein, we develop dual macromolecule-biomineral hybrid species to separately integrate Ex4 and GOx to achieve selective immobilization and release of specific components. In nature, living organisms always produce diverse inorganic–organic complexes to help them smartly respond to various biological stimuli. Inspired by nature, we introduce biomimetic minerals to help construct biocompatible and smart systems[38]. To avoid quick leakage from the delivery system, GOx is immobilized in copper phosphate mineralized particles to form hybrid nano-flowers (mineralized GOx, m-GOx), which are stable under various biological stimuli[39]. Meanwhile, another mineralized particle, calcium phosphate, is applied to integrate Ex4 through in situ biomineralization reaction (mineralized Ex4, m-Ex4), which has been proven to bio-compatibly encapsulate Ex4 in a biomimetic environment based on our previous study[40]. To develop a potent glucose-responsive system (GRS), we combine dual mineralized particles together, as m-GOx may serve as a stable solid enzyme to convert glucose stimuli to $H^+$ signals, and calcium phosphate is a kind of pH-sensitive biomineral that can spontaneously dissolve under acidic conditions[39, 41–45]. To realize facile administration, the dual mineralized particles are loaded onto an alginate-based MN-array patch for convenient and non-invasive drug delivery[28–30]. Importantly, the integration of mineralized particles can significantly improve the mechanical strength of MN for skin penetration by crosslinking effect to reinforce the alginate gel at the cost of reducing the extensibility of the polymeric networks[46]. In this way, a patch-based delivery system providing long-term glucose responsiveness and on-demand Ex4 release is constructed, which may serve as a reliable and effective candidate for glucose regulation in T2D.

## Results

**Design and characterization of m-Ex4 particles**. To fabricate the smart MN-array patch loaded with the separate glucose-sensing element and drug-releasing module to achieve pathology-specific response (Fig. 1), m-Ex4 particles were firstly synthesized as the drug-releasing module. The mineralization of Ex4 proceeded in a biomimetic medium, Dulbecco's Modified Eagle's Medium (DMEM), at a temperature of 37 °C with 5% $CO_2$[40, 42–45]. Like many biomineralization-related proteins, Ex4 (isoelectric point of

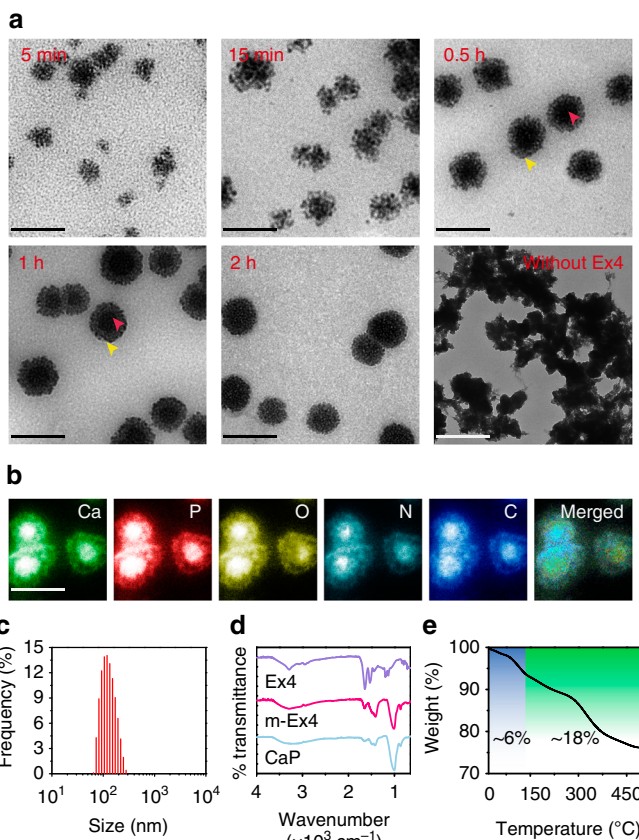

**Fig. 2** Characterization of m-Ex4 nanoparticles. **a** TEM images of m-Ex4 nanoparticles at different time points. The sans peptide group was tested as a control. Red arrows indicate the center; yellow arrows indicate the periphery. Scale bar, 200 nm. **b** Elemental mapping of intermediate at 0.5 h during Ex4 mineralization reaction. Scale bar, 200 nm. **c** Dynamic light scattering (DLS) analysis of m-Ex4 particles. **d** The FTIR spectrum of free Ex4, control calcium phosphate, and m-Ex4 nanoparticles. **e** Thermogravimetric analysis (TGA) of m-Ex4 particles. Blue highlighting indicates the water loss and green highlighting indicates the organic component loss

4.96) contains a high proportion of acidic amino acid residues, including five glutamic acid residues (pKa = 4.25) and one aspartic acid residue (pKa = 3.86) per molecule[47]. These acidic residuals can chelate calcium ions from supersaturated metastable solutions, increasing local supersaturation and providing nucleation sites[48]. After spontaneous nucleation, many nanoscale calcium phosphate crystals were generated around the periphery of the peptide, which tended to form organized spherical structures by means of a bricks-and-mortar self-assembly mechanism[49]. The formation process of m-Ex4 particles in the reaction medium was monitored by transmission electron microscopy (TEM). Initially (5 min), small and loose nano-clusters (around 50–60 nm) that consisted of many tiny crystals (around 2–3 nm) were observed (Fig. 2a), suggesting a quick nucleation and assembly process, consistent with the kinetic analysis (Supplementary Fig. 1). The main elemental composition of the crystals was determined to be Ca, P, O, N and C, suggesting calcium phosphate–peptide complex formation (Fig. 2b). Over time, the clusters progressively became denser (from the periphery to the center) and larger (Fig. 2a, b), and finally, particles around 150 nm were formed (Fig. 2c). These particles were stable in the medium for at least 1 week (Supplementary Fig. 2). Notably, when Ex4 was absent in the medium, only irregular precipitates were generated (Fig. 2a), demonstrating that the organized

structures originated from peptide-mediated mineral generation. Furthermore, Fourier transform infrared (FTIR) spectroscopy of the composites showed the characteristic peaks of inorganic phosphate ($1000–1100$ $cm^{-1}$, P–O vibration) and Ex4 peptide ($1500–1600$ $cm^{-1}$, C=O vibration), confirming the construction of inorganic–organic hybrid structures (Fig. 2d). The organic components accounted for around 18 wt% of the nanoparticles, whereas inorganic calcium phosphate contributed 76% of the total weight, and the content of water was around 6 wt% (Fig. 2e). Collectively, all the data suggested that calcium phosphate-based biomimetic mineralization could serve as a facile and controllable technique to load Ex4, which mimicked the natural biomineralization process to create an inorganic–organic complex, providing a highly biocompatible and biofriendly peptide encapsulation strategy.

**Smart Ex4 release based on dual mineralized particles.** To develop a glucose-responsive delivery system, GOx was introduced in the system as a glucose-sensing element as it rapidly transformed glucose to produce $H^+$ signals[31–34]. To prevent GOx escaping from the delivery system, GOx was immobilized in robust mineralized particles composed of copper phosphate (Fig. 1), to form a mineralized GOx hybrid (m-GOx)[39, 41]. The interaction between amide groups in the protein backbone and copper ions could form abundant complexes, which provided a substrate for nucleation of primary crystals. Large agglomerates of protein molecules and primary crystals were generated, followed by anisotropic growth which resulted in the formation of a branched flower-like structures (Fig. 3a, b)[39, 41]. Elemental mapping analysis indicated that m-GOx mainly contained elements from copper phosphate (Cu, P, O) and protein (N, O) (Fig. 3c), suggesting the successful formation of the inorganic–organic hybrid structure. The organic component contributed about 15 wt% and the inorganic copper phosphate accounted for 75% of the weight (the remaining 10 wt% was water) (Fig. 3d). Importantly, the enzyme activity was negligibly affected after mineralization (detected by tetramethylbenzidine oxidation, Supplementary Fig. 3), and the reactivity was well maintained during storage at room temperature ($25\,°C$) for at least 10 days compared to free enzyme (Fig. 3e), validating long-term glucose responsiveness. This stability could be ascribed to molecular mobility restriction of the entrapped enzyme by mineral particles that acted as physiochemical microanchors[50]. To examine the sensitivity of the glucose-responsive property, m-GOx was incubated with different glucose concentrations, including a hyperglycemic level ($400\,mg\,dl^{-1}$), a normoglycemic level ($100\,mg\,dl^{-1}$), and a control level ($0\,mg\,dl^{-1}$). Notably, glucose oxidation reaction was concentration-dependent. High glucose level induced a greater pH decrease (from 7.4 to 4.6, Fig. 3f) within 2 h, whereas control and normal glucose level only resulted in mild pH change (Fig. 3f). $H_2O_2$ generation showed the same tendency (Fig. 3g), further confirming concentration-mediated reactive fashion. Importantly, during the reaction process, m-GOx was quite stable and robust (Supplementary Fig. 4), demonstrating long-term responsiveness. Moreover, only hyperglycemia-triggered pH decrease (below 6) could lead to the quick dissociation of m-Ex4 particles compared with the other conditions (Fig. 4a), which could be confirmed by TEM observation and DLS measurement (Supplementary Fig. 5). Encouraged by the accurate glucose-sensing and rapid pH-responsive property of the two types of mineralized particles, we combined and encapsulated them in the alginate hydrogel as a potent GRS to achieve a pathology-specific drug release. The release kinetics in response to distinct glucose levels showed that faster Ex4 release was observed in a hyperglycemic environment relative to

control and normoglycemic state (Fig. 4b). With an increase in glucose level, the release rate was significantly elevated (Fig. 4c), suggesting highly accurate glucose concentration-dependent responsiveness. Moreover, when alternatively exposed between a normal and hyperglycemic state every 1 h for several cycles, the GRS containing dual mineralized particles responded to changes in glucose concentrations rapidly for at least five rounds without any responsiveness decrease (Fig. 4d), whereas GRS with free GOx failed to do so (Supplementary Fig. 6). This could be attributed to the immobilization effect of copper phosphate mineralized particles, which were stable under various pH conditions to retard GOx leakage. Meanwhile, calcium phosphate mineralized particles effectively responded to pH decrease and released Ex4, which diffused rapidly from the porous hydrogel structures (Supplementary Fig. 7). Therefore, this combination of two different mineralized particles could provide a highly adjustable and flexible strategy to retain or release specific components with the help of distinct bio-responsiveness of dual mineralized particles. Moreover, the separation of the glucose-sensing element and drug-releasing module allowed for independent regulation of GOx and Ex4, with minimal disruption to the respective loading of the two components. Obviously, lower GOx amount led to slower Ex4 release due to the delayed glucose responsiveness (Fig. 4e). Adjusting Ex4 amount showed less influence on release kinetics (Fig. 4f) but determined the total administered dose. In this way, the release behavior of the GRS could be conveniently controlled and regulated by simply adjusting the GOx amount. In addition, circular dichroism (CD) (Fig. 4g) detection revealed no conformational change of released drug. High-performance liquid chromatography (HPLC) (Fig. 4h) and mass (Supplementary Fig. 8) analysis indicated no degradation occurred and only a small proportion of thiol groups (<5%) was oxidized during the release process, further suggesting mineralization modification and the release process to be highly biocompatible.

**Fabrication and characterization of MN-array patches.** To achieve facile administration, the MN-array patch containing dual mineralized particles was fabricated for convenient and painless treatment (Fig. 1). Briefly, m-GOx and m-Ex4 at optimized concentrations were loaded into the tips of a silicone mold for MNs by vacuum and centrifugation processes, followed by the addition of 2% alginate, which was crosslinked by calcium ions[51]. The produced MNs were arranged in an $11 \times 11$ array with a total area of $6 \times 6$ $mm^2$ (Fig. 5a). Each needle was conically shaped, with a height of 500 μm, base radius of 200 μm, and a tip radius of ~5 μm (Fig. 5b, c). In addition, Cy5-labeled GOx (Cy5-GOx) and FITC-labeled Ex4 (FITC-Ex4) were introduced during the mineralization reaction and MN fabrication processes to help distinguish the two components through the disparity in emission wavelength between the two dyes (Supplementary Fig. 9). Evidently, red and green emissions were concurrently observed in every needle (Fig. 5d), demonstrating the uniform integration of the two compounds across the whole patch. In each needle, red and green signals rarely overlapped (Fig. 5e), confirming separate integration. Notably, the fabricated MN patch was stable and robust without any curling or collapse during long-term storage ($4\,°C$, 30 days, Fig. 5f), and measurement of mechanical strength using a tensile compression apparatus indicated that integration of mineralized particles significantly increased the failure force of blank alginate MNs by 1.93 fold (from 0.15 to 0.29 N per needle) (Fig. 5g), owing to the crosslinking effect of mineralized particles, which reinforced the stability of the MN at the cost of reducing the extensibility of the polymer network[46]. Importantly, this range of failure force (≥16 N per array)[52] indicated sufficient MN stiffness and strength to facilitate skin penetration without

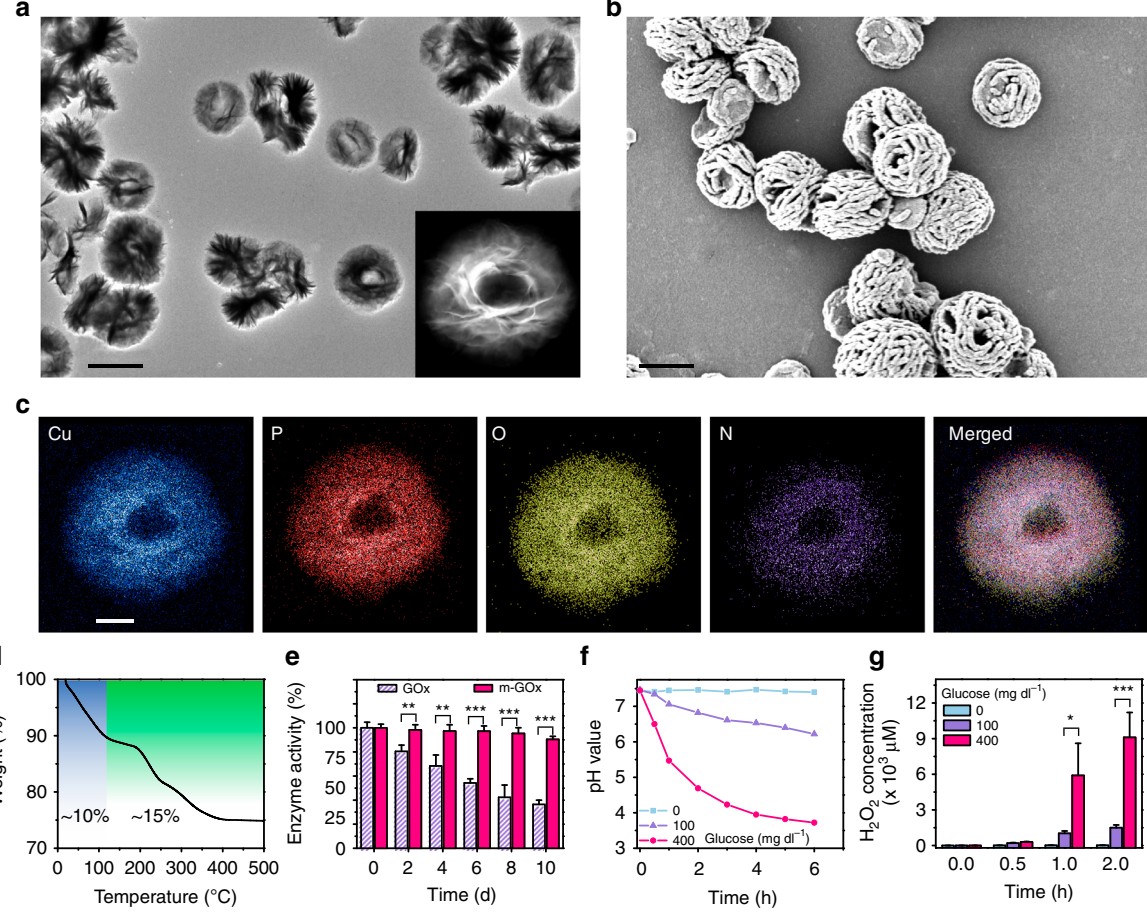

**Fig. 3** Characterization of m-GOx particles and glucose responsiveness. **a** TEM images of m-GOx. Magnification is the single m-GOx particle in dark field. Scale bar, 3 μm. **b** Scanning electron microscopy (SEM) image of m-GOx particles. Scale bar, 2 μm. **c** Elemental mapping of m-GOx particles. Scale bar, 1 μm. **d** TGA of m-GOx particles. Blue highlight indicates the water loss, and green highlighting indicates the organic component loss. **e** Enzyme activity decay of free GOx vs. m-GOx over time. Mean ± S.D. ($n = 3$). **$P < 0.01$, ***$P < 0.001$ (two-tailed Student's $t$-test). **f** Relevant pH decrease of different incubation solutions with the m-GOx particles. **g** Relevant $H_2O_2$ generation of different incubation solutions with the m-GOx particles. Mean ± S.D. ($n = 4$). *$P < 0.05$, ***$P < 0.001$ (two-tailed Student's $t$-test)

deformation[53]. Moreover, after exposure to glucose solution, MNs maintained the original morphology and shape (Supplementary Fig. 10), serving as a stable support to guarantee long-term administration. Meanwhile, obvious FTIC-Ex4 release and Cy5-GOx retention were observed after glucose exposure (Fig. 5h), further verifying our hypothesis. In this regard, a well-designed glucose-sensitive Ex4 delivery device based on mineralized particles and alginate MN arrays was successfully achieved, which may potentially offer non-invasive and on-demand Ex4 delivery for blood glucose regulation.

**In vivo studies of the MN-array patches for T2D therapy.** To evaluate the in vivo antidiabetic effect of MN-array patches for T2D therapy, C57BL/6 db/db mice were grouped and transcutaneously exposed to different samples, including free Ex4, MN/Ex4, MN/m-Ex4, m-Ex4/m-GOx, MN/m-Ex4/GOx and MN/m-Ex4/m-GOx. The loading amount of Ex4 and GOx in each MN patch could be regulated based on the input during fabrication. When compared to free Ex4, the amount of Ex4 in each group was set to around 20 μg. As shown in Fig. 6a, the MN-array patch effectively penetrated dorsum skin of the mouse to form microchannels, which could be observed with help of trypan blue (Fig. 6a) and hematoxylin & eosin (H&E) staining (Supplementary Fig. 11). Moreover, the channels recovered within 4 h (Supplementary Fig. 12), and there was no obvious damage

observed in the region following recovery (Supplementary Fig. 13). Interestingly, MN patch administration induced a relatively mild inflammation reaction (IL-1α, IL-1β, IL-6, and TNF-α expression, Supplementary Fig. 14) compared to intradermal (ID) injection of dual mineralized particles, mainly due to less invasiveness of the MN patch to the skin, suggesting a highly biocompatible treatment. The blood glucose level (BGL) of treated mice in each group was monitored over time (Supplementary Fig. 15). Obviously, minimal fluctuations of BGL (~401.8 ± 15.6 to 453.3 ± 31.2 mg dl⁻¹) were detected in the control group (Fig. 6b), indicating the successful establishment of a stable diabetic model. When free Ex4 was administered, the BGL of diabetic mice was decreased by 66.6% (from about 443.2 to 148.1 mg dl⁻¹) within 6 h, but the normoglycemic state could not be maintained, and returned to a hyperglycemic state within 24 h, likely due to rapid drug clearance. Simple MN encapsulation of free Ex4 could not provide much help as the free drug quickly leaked out from the porous network of alginate-based MN (Supplementary Fig. 7). Besides, singular m-Ex4 integration in MNs exhibited poor BGL regulation due to the lack of a glucose-sensing module, rendering drug release ineffective. In contrast, dual mineralized particle-containing MN patches (MN/m-Ex4/m-GOx) showed promising BGL control. Even after 72 h treatment, BGL in treated mice was still less than the original level (Fig. 6b, c), owing to the smart glucose responsiveness of m-GOx

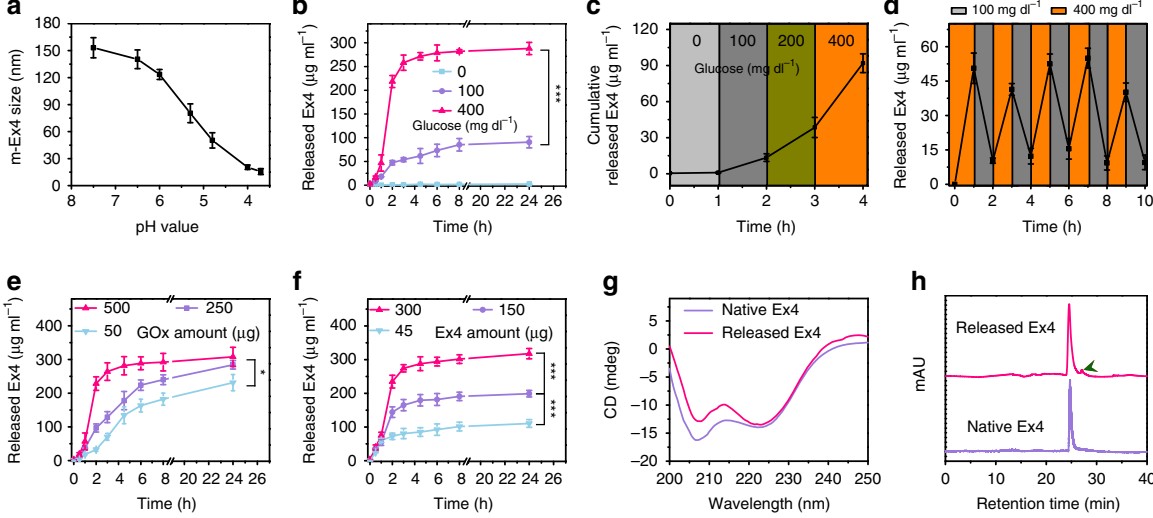

**Fig. 4** In vitro glucose responsiveness of dual mineralized particle-based GRS. **a** m-Ex4 size change in response to pH. Mean ± S.D. ($n = 3$). **b** Accumulated Ex4 release from dual mineralized particle-based GRS in different glucose concentrations at 37 °C. Mean ± S.D. ($n = 3$). ***$P < 0.001$ (two-tailed Student's $t$-test). **c** The increase of glucose concentration significantly elevated the Ex4 release rate. Mean ± S.D. ($n = 3$). **d** The pulsatile release profile of dual mineralized particle-based GRS presented the concentration of released Ex4 as a function of glucose concentration (100 and 400 mg dl$^{-1}$). Mean ± S.D. ($n = 3$). **e** Accumulated Ex4 release from dual mineralized particle-based GRS with different amounts of GOx at 37 °C. Mean ± S.D. ($n = 3$). *$P < 0.05$ (two-tailed Student's $t$-test). **f** Accumulated Ex4 release from dual mineralized particle-based GRS with different amounts of Ex4 at 37 °C. Mean ± S.D. ($n = 3$). ***$P < 0.001$ (two-tailed Student's $t$-test). **g** CD spectrum of released and native Ex4. **h** HPLC analysis of released and native Ex4. Retention time is ~25 min. Arrow indicates the peak from the oxidative product of Ex4 according to the mass analysis

and rapid dissociation of m-Ex4. It should be mentioned that dual mineralized particles without MN could only provide a transient responsiveness because the two particles were easily separated in the different locations of intradermal spaces, and relatively high inflammation also contributed to the response failure (Supplementary Fig. 14). Therefore, MN and dual mineralized particles were both essential in the system. Notably, the immobilization of glucose-sensing element played a significant role, with evidences that free GOx could not ensure long-term glucose responsiveness (Fig. 6b, c) due to its rapid escape from the system. This phenomenon could be further confirmed by the intraperitoneal glucose tolerance test (IPGTT). Clearly, after 1 h treatment, both MN/m-Ex4/GOx and MN/m-Ex4/m-GOx exhibited significant potential to inhibit rapid BGL increase, suggesting effective responsiveness (Fig. 6d, f). However, after 12 h treatments, MN/m-Ex4/GOx lost BGL control capability while MN/m-Ex4/m-GOx was still effective (Fig. 6e, f). The escape of free GOx and the retention of m-GOx in MN after treatment could be directly observed based on fluorescence signals (Supplementary Fig. 16), further confirming the hypothesis. Plasma Ex4 measurement by enzyme-linked immunosorbent assay (ELISA) revealed the continuous replenishment of the drug under MN/m-Ex4/m-GOx patch treatment (Fig. 6g, h), which was much improved compared to other treatments, consistent with BGL detection. To extend the normoglycemia duration (<150 mg dl$^{-1}$) time, the Ex4 amount could be precisely increased. In our study, around 300 μg Ex4 per patch was set as the up limit as a higher dose induced certain cavity block and uneven particle distribution in the backing layer during MN fabrication. Obviously, greater Ex4 amount resulted in prolonged normoglycemia duration (Fig. 6i; Supplementary Fig. 17). The 300 μg Ex4-loaded patch administration provided 5 days normoglycemia maintenance and a slow BGL increase to the original level (9 day), which is somewhat better than existing long-acting formulations[54–58]. Moreover, lower GOx amount reduced the glucose responsiveness of the patch, which might markedly decrease the BGL regulatory capacity of the patch (Fig. 6j; Supplementary Fig. 17), thus

adequate GOx (>500 μg) was necessary to maximize therapeutic performance. In addition, as opposed insulin, continuous Ex4 treatment did not increase the risk of hypoglycemia when administered in a normoglycemic state (Fig. 6k). This was consistent with the corresponding hypoglycemia index (defined as the fall in glucose from the initial reading to the nadir divided by the time over which this fall was reached) (Fig. 6l), ensuring the safety of continuous Ex4 exposure. In summary, all the observations demonstrated that the MN-array patch separately containing dual mineralized particles could serve as a smart and potential candidate to improve Ex4 therapeutic performance. This strategy revealed that isolating the glucose-sensing element and drug-releasing module could ensure long-term responsiveness and smart drug release. We believe this well-designed MN-array patch can achieve ideal self-regulated and on-demand therapy for T2D patients in future.

## Discussion

Currently, GOx-based glucose-responsive delivery systems mainly employ matrices consisting of pH, hypoxia, or H$_2$O$_2$-sensitive materials, which release antidiabetic drugs by either dissociation or degradation in response to microenvironmental changes[28–34]. However, GOx and antidiabetic drugs are frequently encapsulated into the same formulation, which may release the entire payloads during the disassembly process[21–23]. In this way, the effectiveness of multi-round glucose responsiveness is doubtful[35]. To specifically release Ex4 while immobilizing GOx, two types of mineralized particles (copper phosphate and calcium phosphate) with different responsiveness to bioenvironmental stimuli were introduced. The robust copper phosphate could avoid the undesired release of GOx and maintain its activity for a long time. Meanwhile, calcium phosphate responded sensitively to pH decrease induced by glucose oxidation, leading to rapid Ex4 release. In this way, a long-term GRS was achieved. In addition, the specific Ca-based particle dissolution and Cu-based particle retention induced a mild fluctuation of blood Ca level without noticeable Cu level increase

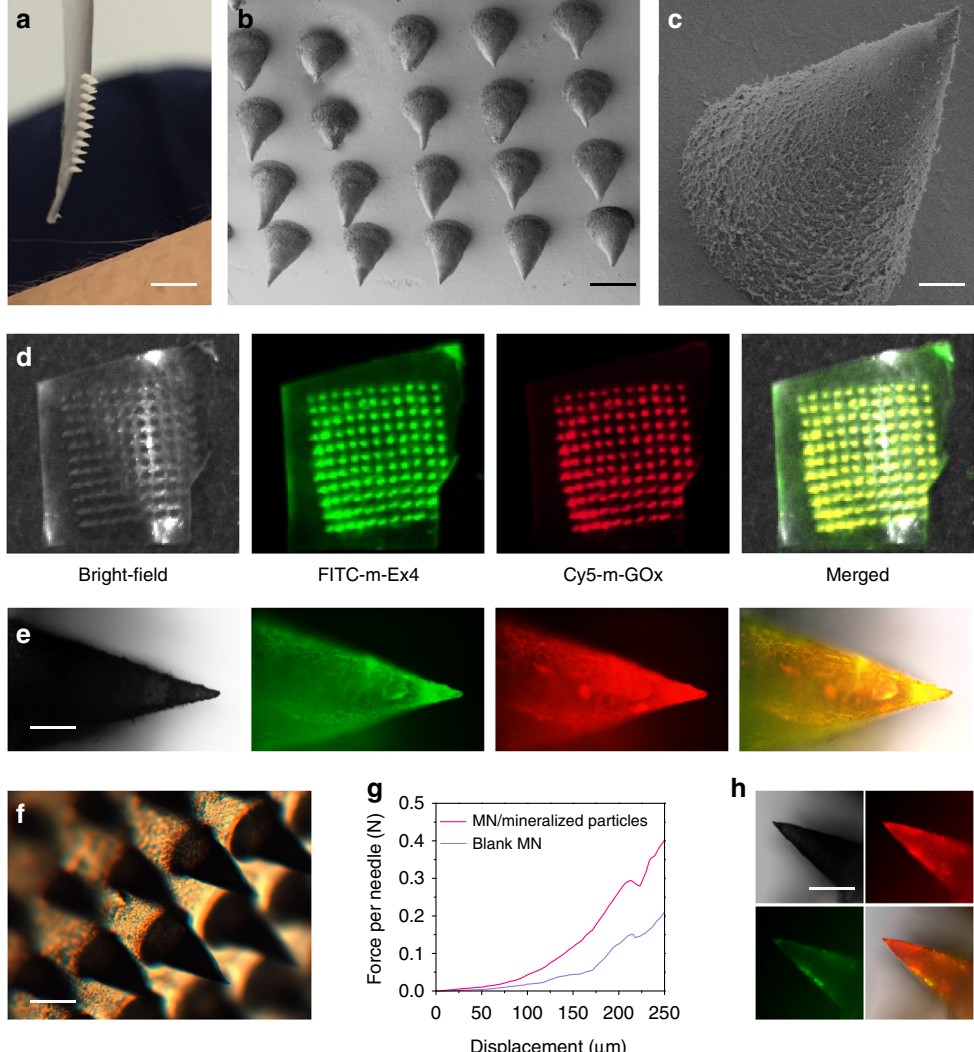

**Fig. 5** Characterization of dual mineralized particle-loaded MN-array patches. **a** Photograph of the smart Ex4 patch with an MN array. Scale bar, 0.5 cm. **b** SEM image of the MN array. Scale bar, 500 μm. **c** Magnification of a single microneedle. Scale bar, 100 μm. **d**, **e** Fluorescence microscopy images of the MN arrays and a single needle containing FITC-m-Ex4 and Cy5-m-GOx. Scale bar, 150 μm. **f** Bright-field image of the MN array after 30 days storage. Scale bar, 400 μm. **g** Mechanical behaviors of MNs with or without mineralized particles. **h** Fluorescence images of a single microneedle containing FITC-m-Ex4 and Cy5-m-GOx after exposure to 400 mg dl$^{-1}$ glucose for 24 h. Scale bar, 300 μm

(Supplementary Fig. 18). The phenomenon indicated negligible blood ion-based side effects, suggesting high biocompatibility and safety of our dual mineralized particle formula. To realize less invasive (more facile) administration, the dual mineralized particles were incorporated into the indissoluble MN-array patch. Interestingly, the incorporation of mineralized particles significantly improved mechanical strength and skin penetration capability of the patch but had little influence on the diffusion of small molecules from porous gel network[46], demonstrating a sustained transdermal administration. Compared to traditional ID injection, MN-array patch treatment induced a milder inflammatory reaction due to less invasiveness of the patch approach, and after treatment, the patch could be easily removed, suggesting a disposable manner. Considering the bio-sourced materials are quite low-cost and biocompatible, we believe the well-designed MN-array patch can serve as an economic, convenience, safe, and highly effective candidate for long-term T2D treatments.

To apply the technique from bench to clinic, there are still some challenges that should be addressed. Firstly, in the clinic, T2D patients are typically treated by Ex4 twice a day before the first and last meal, with the dose of 5 or 10 μg[13]. The high treatment frequency has provoked the desire to develop long-acting release (LAR) Ex4 formulations[54, 59]. One of the most successful LAR systems is BYDUREON®, which contains 2 mg Ex4 and can be administered once per week[54, 59]. Although our 300 μg Ex4-loaded patch also shows promising BGL control for almost 1 week (5 to 6 days) in the mouse model, we believe this dosage needs to be raised in human cases due to the differences in pharmacokinetics and pharmacodynamics between the mouse and the human[60]. Therefore, a longer MN is under development to predictably increase the Ex4 loading amount, which will be more suitable for clinical applications. Secondly, the frequency for applying MN-array patches is another issue as the current study has not established a stable blood drug level for long-term therapy. Repeated treatments by Ex4 formulations have proven to be able to induce steady therapeutic drug concentration (0.1–0.3 ng ml$^{-1}$) in T2D patients after several weeks[54, 59]. Moreover, it has been confirmed that repeated MN-array patch treatments do not alter skin appearance or barrier function, and cause no measurable disturbance of serum biomarkers of infection, inflammation, or immunity, suggesting high biocompatibility of the microneedle

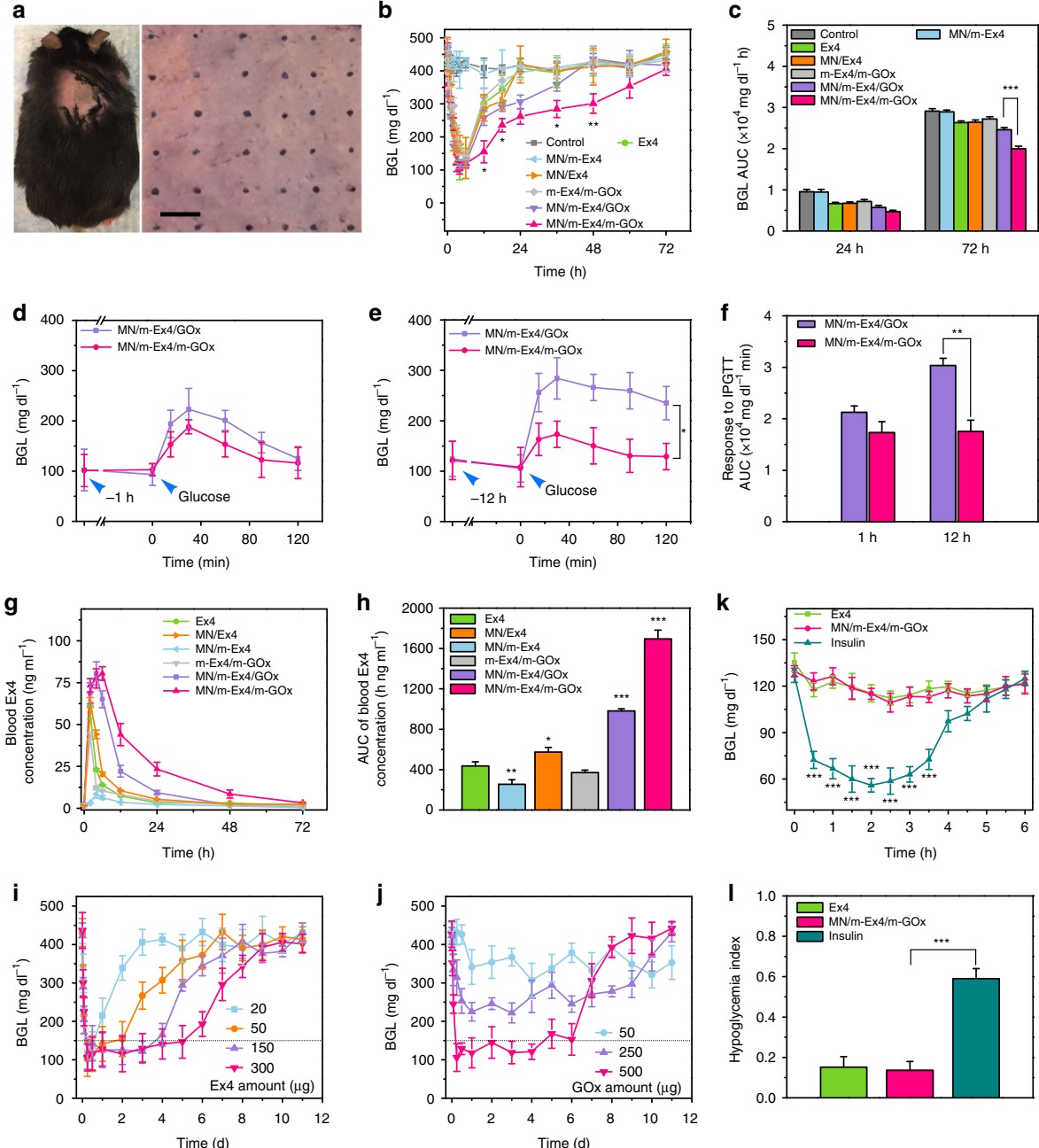

**Fig. 6** In vivo studies of the MN-array patches for T2D therapy. **a** Photograph of the mouse treated by an MN patch at the dorsum; trypan blue staining showing microchannels after MN patch treatment. Scale bar, 500 μm. **b** Long-term BGL monitoring in C57BL/6 db/db mice under different treatments. Mean ± S.D. ($n = 3$). *$P < 0.05$, **$P < 0.01$ for MN/m-Ex4/m-GOx treated group compared with MN/m-Ex4/GOx treated group (two-tailed Student's t-test). **c** The area under the curve (AUC) of BGL integrated up to 24 and 72 h. Mean ± S.D. ($n = 3$). ***$P < 0.001$ (two-tailed Student's t-test). IPGTT of diabetic mice at 1 h **d** and 12 h **e** post-administration of MN/m-Ex4/GOx and MN/m-Ex4/m-GOx. Mean ± S.D. ($n = 3$). *$P < 0.05$ (two-tailed Student's t-test). **f** Responsiveness to IPGTT was calculated based on the AUC over 120 min. Mean ± S.D. ($n = 3$) **$P < 0.01$, (two-tailed Student's t-test). **g** Blood drug concentration vs. time curves of different treatments. Mean ± S.D. ($n = 3$). **h** AUC of blood drug concentration up to 72 h. Mean ± S.D. ($n = 3$) *$P < 0.05$, **$P < 0.01$, ***$P < 0.001$ for administration with different formulations compared with Ex4 (two-tailed Student's t-test). **i** Long-term BGL monitoring in C57BL/6 db/db mice under MN/m-Ex4/m-GOx administration with different Ex4 amounts. Mean ± S.D. ($n = 3$). **j** Long-term BGL monitoring in C57BL/6 db/db mice under MN/m-Ex4/m-GOx administrations with different GOx amounts. Mean ± S.D. ($n = 3$). **k** Blood glucose changes of healthy mice administered with Ex4, MN/m-Ex4/m-GOx, and insulin. Mean ± S.D. ($n = 3$). ***$P < 0.001$ for insulin treated group compared with MN/m-Ex4/m-GOx treated group (two-tailed Student's t-test). **l** Quantification of the hypoglycemia index, which was calculated from the difference between the initial and nadir blood glucose readings divided by the time at which nadir was reached. Mean ± S.D. ($n = 3$). ***$P < 0.001$ (two-tailed Student's t-test)

platform[61]. Hence, in the next step, we plan to optimize administration frequency to establish a stable therapeutic blood Ex4 concentration in large animal models for clinical translation of the MN-array patch-based strategy. The third challenge lies in

the difference between the thickness of mouse and human skins. The human skin has relatively thicker epidermal (130–180 μm) and dermal (~2000 μm) layers compared to the mouse skin[62]. In addition, the mouse skin has densely packed hair follicles,

whereas the human skin has larger areas of interfollicular skin with sparse hair follicles[62]. Thereby, we will attempt to optimize the length, shape, and morphology of MNs to translate the designed MN-array patch from the mouse into the human in future.

In summary, we successfully developed a potent alginate-based MN-array patch, which contained dual mineralized particles for painless and convenient Ex4 treatment. Different from other approaches, this patch separately encapsulated a robust glucose-sensing element and a sensitive drug-releasing module to avoid rapid loss of the glucose-sensitive component during the drug release process. The patch exhibited promising glucose-regulation capability by a specific response to hyperglycemic state with rapid onset, suggesting a pathology-specific regulation. After inducing a normal glucose level, release rate spontaneously slowed down, demonstrating a leverage-based regulatory strategy. In this way, a closed-loop and feedback-controlled Ex4 treatment for type 2 diabetes was achieved. As for biocompatibility of device, the framework of the MNs, alginate, has been previously approved by the FDA[63]. Moreover, the biomineral calcium phosphate is native to our body as it is the main component of biological hard tissues such as bone and tooth[64]. The decomposition products during Ex4 release are mainly inorganic ions (e.g., calcium and phosphate ions), which act as nutrients to be absorbed by surrounding biological tissues. In this manner, a safe and smart artificial device that can be intelligently activated and self-regulated in response to pathology signals can serve as a highly potent candidate for diabetes therapy.

## Methods

**Preparation of m-Ex4.** Briefly, 2 mg Ex4 was dissolved directly in 1 ml DMEM medium for 24 h to reach the equilibrium. Overall, 10 μl CaCl₂ (1 M) was introduced into the reaction for incubation at 37 °C under 5% CO₂ for another 24 h. After incubation, the mineralized Ex4 was purified by centrifugal ultrafiltration (100k MWCO, 6000 rpm, 20 min). The solid was lyophilized and stored at −20 °C until use.

**Preparation of m-GOx.** Briefly, 3 mg GOx was dissolved directly in 6 ml PBS for 0.5 h to reach the equilibrium. A total of 40 μl molecular-biology-grade CuSO₄ (120 mM) was added to GOx solution, followed by incubation at 4 °C for three days. The nanoflower precipitate was collected, washed with deionized water and dried under vacuum. The protein concentration in the supernatant was measured by Pierce BCA Protein Assay Kit (Thermo Fisher Scientific, USA) using BSA as a standard.

**In vitro drug release measurements.** After preparation of m-GOx particles, various glucose solutions (0, 100, or 400 mg dl⁻¹ in PBS, 500 μl) were added to each tube and incubated at 37 °C for a period of time. At each time point, the pH values of supernatant were measured by HI 2210 pH meter (HANNA instruments, USA). After 6 h incubation, the supernatant was collected by centrifugation (10,000 rpm, 10 min), followed by addition to m-Ex4 particles. After another 2 h incubation, the samples were collected for TEM observation and DLS analysis under a Philips/FEI CM200 Microscope (USA) and an SZ-100 Nanoparticle Analyzer (HORIBA Scientific, Japan), respectively.

To detect the released drug, the dual mineralized particles containing around 300 μg Ex4 and 500 μg GOx were integrated with Ca²⁺-crosslinked alginate hydrogel (0.3 ml). The combination systems were exposed to different concentrated glucose solutions (1 ml) for a period of time. At each time point, a 50 μl supernatant was collected and the Ex4 concentration was determined by Ex4 EIA Kit (Phoenix Biotech, USA, catalog number: EK-070–94) according to the standard protocol. Moreover, to evaluate the responsiveness of the GRS to changes in glucose levels, the system was exposed to glucose concentrations which increased from 100 to 400 mg dl⁻¹, or periodically switched between 100 and 400 mg dl⁻¹ at 1 h intervals. The released amount of Ex4 was measured by EIA kit (Phoenix Biotech, USA).

To access the influence of Ex4 and GOx amount on the release kinetics of the GRS, mineralized particles containing 300 μg Ex4 and different amounts of GOx (500, 250, or 50 μg), or 500 μg GOx and different amounts of Ex4 (300, 150, 45 μg) were integrated with Ca²⁺-crosslinked alginate hydrogel gel (0.3 ml). The Ex4 release detection was according to the aforementioned method. After drug release, a Perkin-Elmer 200 series HPLC system with a Waters 2487 UV detector and a Bioscan Flow-Count detector equipped with an analytical C-18 HPLC column (XTerra 5 μm, 150 × 4.6 mm, Waters) was used for analyzing the released Ex4. HPLC

conducted under a stepwise gradient from 10 to 90% of Buffer B (0.1% TFA in water) within 30 min (Buffer A was 0.1% TFA in acetonitrile). Liquid chromatography–mass spectrometry (LC–MS) analysis was conducted on a Waters LC–MS system (Waters, Milford, MA, USA) that included an Acquity HPLC unit coupled to the Waters Q-Tof Premier high-resolution mass spectrometer. An Acquity BEH Shield RP18 column (150 × 2.1 mm) was employed for chromatography. The entire column elute was introduced into the Q-Tof mass spectrometer. Ion detection was achieved in ESI mode using a source capillary voltage of 3.5 kV, source temperature of 110 °C, desolvation temperature of 200 °C, cone gas flow of 50 l h⁻¹ (N₂), and desolvation gas flow of 700 l h⁻¹ (N₂). Moreover, aliquoted samples were analyzed on a J-810–150S CD spectrometer (JASCO, Japan) for conformational analysis.

**Fabrication of dual mineralized particle-loaded MNs.** All of the MNs in this study were fabricated using six uniform silicone molds from Blueacre Technology Ltd (Dundalk, Ireland). Following the preparation of dual mineralized particles, m-GOx (containing 0.5 mg GOx) was firstly evenly dispersed in 300 μl of PBS. The suspension was directly deposited by pipette onto the MN mold surface (50 μl per array). Molds were then placed under vacuum (400 mmHg) for 15 min to allow the m-GOx suspension to flow into the MN cavities. The covered molds were then transferred to a Sorvall ST 8 Small Benchtop Centrifuge (Thermo Scientific, USA) for 25 min at 2500 rpm to compact m-GOx into MN cavities. The process was repeated six times and m-Ex4 was loaded in MN cavities by a similar process. To improve MN morphology, a piece of adhesive tape (3/4 × 4-inch) was applied around the 1.5 × 2-cm micromold baseplate. Then 3 ml of alginate (2%) was added to the prepared micromold reservoir followed by centrifugation (2500 rpm, 25 min). Finally, 150 μl CaCl₂ (1 M) was added to the micromold reservoir followed by drying at 25 °C for 24 h. After the desiccation was completed, the MN-array patches were carefully separated from the silicone mold. The resulting product could be stored at 4 °C in a sealed six-well container for at least 30 days. The morphology of the MN arrays was observed on a Hitachi SU-70 Schottky field emission gun SEM. Moreover, FITC-labeled Ex4 and Cy5-labeled GOx were used during the mineralization reaction and MN fabrication. The fluorescence signal of fabricated MN arrays was investigated under an Olympus IX81 inverted fluorescence microscopes (Japan) and Maestro all-optical imaging system (Caliper Life Sciences, Hopkinton, MA, USA), respectively.

**Mechanical strength test.** The mechanical strength of the MN arrays with or without mineralized particles was measured by pressing MN arrays against a stainless-steel plate on an MTS 30 G tensile testing machine. The initial gauge was set as 2 mm between the MN tips and the stainless-steel plate, with 10.00 N as the load cell capacity. The speed of the top stainless-steel plate movement toward the MN arrays was set as 0.1 mm s⁻¹. The failure force of MNs was recorded as the needle began to buckle.

**Skin penetration efficiency test.** After MN arrays were applied to the dorsum of the mouse skin for 1 h, the treated skin was stained with trypan blue for 30 min before wiping residual dye from the skin surface with tissue paper. The mouse was euthanized and the skin sample was viewed under an Olympus BX41 optical microscope (Japan).

**In vivo studies using C57BL/6 db/db mice.** Glucose-lowering capabilities of MN-array patches were evaluated in C57BL/6 db/db mice (6–8 weeks, male, 40–50 g), which were purchased from Harlan Laboratories. All animal studies were conducted in accordance with the principles and procedures outlined in the Guide for the Care and Use of Laboratory Animals and approved by the Institutional Animal Care and Use Committee (IACUC) of the Clinical Center, National Institutes of Health (the approved number is NIBIB 15–01). Animals were housed under a 12 h light/dark circle, allowed food and water ad libitum, and acclimatized for 2 weeks. The plasma-equivalent glucose was measured from tail vein blood samples (~5 μl) of mice using a True-Track glucose meter (CVS Health, USA). All mice were fasted overnight before administration and divided into seven random groups. Three mice for each group were transcutaneously treated with free Ex4, MN/Ex4, MN/m-Ex4, m-Ex4/m-GOx, MN/m-Ex4/GOx, or MN/m-Ex4/m-GOx. Untreated mice were set as the control group. The groups were not blinded. The amount of Ex4 (0–300 μg) and GOx (0–500 μg) could be regulated based on the loading amount. When comparing with free Ex4, the Ex4 amount was set to 20 μg. The BGL of each mouse was monitored at different time points until a return to the stable hyperglycemia. A glucose tolerance test was conducted to access the in vivo glucose responsiveness of MN arrays. Briefly, C57BL/6 db/db mice (6–8 weeks, male) were fasted overnight and administered MN/m-Ex4/GOx and MN/m-Ex4/ m-GOx with an Ex4 dose of 50 nmol kg⁻¹ for each mouse. After 1 h or 12 h, a glucose solution in PBS was then injected i.p. into all mice at a dose of 1.5 g kg⁻¹. The blood glucose level was monitored at 0, 15, 30, 60, 90, 120 min after injection. Moreover, healthy mice (FVB mice, 4–5 weeks, male, 18–20 g) were used to assess hypoglycemia and were administered Ex4 or MN/m-Ex4/m-GOx, but were not subjected to glucose challenge. Insulin (10 mg kg⁻¹) was used as a positive control. To measure the plasma Ex4 concentration in vivo, a blood sample of 25 μl was

drawn from the orbital sinus at different time points and Ex4 concentration was determined by a commercial Ex4 EIA kit (Phoenix Biotech, USA).

**Data availability**. The authors declare that the data supporting the findings of this study are available within the article and its supplementary information files or from the corresponding authors on reasonable request.

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

## Acknowledgements

This work was supported in part by the China Postdoctoral Science Foundation (2015M570745 and 2016T90815), Natural Science Foundation of Guangdong Province (2016A030310229), National Natural Science Foundation of China (81602634), and the Intramural Research Program, National Institute of Biomedical Imaging and Bioengineering, National Institutes of Health (ZIA EB000073).

## Author contributions

W.C., L.F., and X.C. conceived the project and designed the experiments. W.C., R.T., C.X., B.C.Y., G.W., Y.L., Q.N., F.Z., Z.Z., J.W., G.N., and Y.M. were primarily responsible for the data collection and analysis. W.C. and X.C. prepared the figures and wrote the main manuscript text. All authors contributed to the discussions and manuscript preparation.

## Additional information

**Competing interests:** The authors declare no competing financial interests.

