## [Peer Review File · Nature Communications]

Reviewers' comments:

Reviewer #1 (Remarks to the Author):

The manuscript describe about microneedle-array patches loaded with mineralized encapsulation of peptide (Ex4) with GOx-based glucose responsive sustained release. This might be interesting topic to the readers of Nature Communication. However, it needs major revisions to match the standards of Nature Communication.

Major revisions

1. To prove the novelty of this research, authors need more explanation compared to previous studies using mineralized encapsulation or GOx-based responsive release. Authors should clearly explain new points of this research.
2. To prove the impact of this research, authors need to compare other commercialized technology such as real-time glucose level-responsive releasing device of Nature, 485, S6–S8 (17 May 2012). They should quantitatively explain the pros and cons compared to this recent technology. For example, loading efficiency, loading density, sustained release time, administration frequency, in vivo half-life of particles and stability, etc.
3. To prove the biosafety of this device, authors need to do more biosafety experiments because high concentration of copper ion and H₂O₂ and low pH could be toxic to human. Repeated use could increase copper ion concentration in the body. Therefore, it needs long-term exposure safety check of this device.
4. And, biological indicators should be checked to prove no inflammation such as levels of cytokines or neutrophils with real-time PCR or ELISA.
5. To improve the readability, the manuscript should be totally revised. Sentences and paragraphs are too long. Concise and meaningful English expression should be matched to the standard of Nature Communication.
6. Figures also should become concise. Authors need to choose meaningful quantitative data with minimum qualitative pictures.
7. To prove the stability of device, authors needs to show long-term stability of GOx or Ex-4.
8. In addition, no change of chemical structure should be checked by Mass Spectroscopy instead to HPLC.
9. Authors need to include IACUC approval number for animal experiments.
10. All quantitative experiments need repeat number and statistical analysis to prove the significant difference with method name.

Minor revisions

1. Please change the reference No.2 from website to journals because website can be easily changed.

Reviewer #2 (Remarks to the Author):

This paper provides proof of concept data on the clinical need for better ways of therapeutically treating type 2 diabetes, which is an enormous global healthcare burden. Specifically, the authors propose a method that will reduce the number of injections required which should lead to improved patient compliance and reduced healthcare costs. Significantly, the authors combine a novel slow release therapeutic method with a smart delivery system that operates in a stimulus responsive manner (the release of Ex4 in a glucose responsive fashion). The potential of this strategy is shown in an appropriate small animal model. Closed-loop stimulus-responsive systems have been previously highlighted as a potential application of microneedle technology, making the concept in itself not original; however, there has been a lack of examples reduced to practice where this has been practically achieved and I believe that this paper will be of strong interest to researchers with broad interests in drug delivery.

The manuscript is well written and the experiments are well designed and the results clearly laid-out.

Major Comments

The authors propose that a microneedle system could be used as a smart glucose responsive system, where the release of Ex4 would help maintain glycemic control.

However, for clinical translation, it is unclear from the manuscript what therapeutic dose of Ex4 will be required and if this could be successfully loaded into a microneedle system i.e. with the loading efficiency in their system would they be able to achieve a therapeutically relevant dose in human and what size would the patch be? Are there any limitations of the animal model where the thickness of skin and proximity to blood glucose could impact on the efficacy of this system in clinical use? This information should be added to the discussion

Control of injection under skin? Without MN

Other Comments

The schematic of figure 1 is not immediately clear and could be enhanced

Supplementary Figure S10 – the figure legend should provide additional detail

The authors mention FDA approval for alginate products. References for the most relevant products should be included

Point-by-point Response

Reviewer #1:

Major revisions

1. Comments: *“To prove the novelty of this research, authors need more explanation compared to previous studies using mineralized encapsulation or GOx-based responsive release. Authors should clearly explain new points of this research.”*

Response: Thanks for the comment. We have provided more explanation compared to previous studies including mineralized encapsulation (Ref #42), GOx-based systems (Ref #30-36), MN-array patch-based systems (Ref #30-32, 49-51). We highlight our new points: (i) new dual mineralized particles as GRS; (ii) isolation of glucose-sensing element and drug-releasing module for multi-round responsiveness; (iii) indissoluble alginate MN-array patch rather than dissoluble systems as stable support to guarantee long-term administration. We added more details and descriptions in the text; please see highlights on page 4-7.

2. Comments: *“To prove the impact of this research, authors need to compare other commercialized technology such as real-time glucose level-responsive releasing device of Nature, 485, S6–S8 (17 May 2012). They should quantitatively explain the pros and cons compared to this recent technology. For example, loading efficiency, loading density, sustained release time, administration frequency, in vivo half-life of particles and stability, etc”*

Response: Thanks for the comment. We have compared the other commercialized technology such as electronic/mechanical devices and pointed out their drawbacks including “high cost of the instruments, complicated device management, patient’s discomfort and high susceptibility to biofouling”. (Ref #19-22) Our MN-array patch may avoid the limitations by introducing economic bio-sourced materials and acting in a “throw-away” administration manner. Please see the highlights on Page 4-5. Moreover, we have also compared our patch with other related technology (Ref #30-36, 49-51) and discussed the pros and cons. Please see the highlights on Page 4-7, 26. Compared to commercialized long-acting release (LAR) formulation of Ex4 (Ref #59, 64), our patch is also complete by providing 5-day normoglycemia maintenance and a slow BGL increase to original levels (9 days). Some quantitative data are still under investigation as there are significant differences in pharmacokinetics and pharmacodynamics between mice and human (Ref #65). Optimizing

the dose and administration frequency in primate model is a good choice in our future studies. We have added more discussion in the text; please see highlights on Page 26.

3. Comments: *“To prove the biosafety of this device, authors need to do more biosafety experiments because high concentration of copper ion and H₂O₂ and low pH could be toxic to human. Repeated use could increase copper ion concentration in the body. Therefore, it needs long-term exposure safety check of this device.”*

Response: According to the comment, we have added more biosafety experiments including proinflammatory cytokine gene expression evaluation, and blood Ca and Cu concentration detection. The inflammation induced by the patch treatment is acceptable compared with intradermal (ID) injection (Supplementary Fig. 14). Local H₂O₂ generation did not induce much damage to the dermal tissues according to Supplementary Fig. 13. Moreover, the specific Ca-based particle dissolution and Cu-based particle retention induced a mild fluctuation of blood Ca level without any Cu level increase (Supplementary Fig. 18), indicating high biocompatibility of the system. The mice are all healthy after treatments, and long-term safety evaluation is still underway. We have added the discussion in the text; please see highlights on Page 19, 25

4. Comments: *“And, biological indicators should be checked to prove no inflammation such as levels of cytokines or neutrophils with real-time PCR or ELISA.”*

Response: Thanks for the suggestion. We have evaluated the inflammatory cytokines such as L-1 α , IL-1 β , IL-6 and TNF- α by using qRT-PCR. We found that the MN-array patch induced acceptable inflammation compared to ID injection. Please see the highlights on page 19 and Supplementary Fig. 14.

5. Comments: *“To improve the readability, the manuscript should be totally revised. Sentences and paragraphs are too long. Concise and meaningful English expression should be matched to the standard of Nature Communication”*

Response: Thanks for the suggestion. We have asked a native English speaker to help improve the readability.

6. Comments: *“Figures also should become concise. Authors need to choose meaningful quantitative data with minimum qualitative pictures.”*

Response: Thanks for the comment. We have added and improved some quantitative data. Please see Fig. 3e, Fig 3g, Supplementary Fig. 3, Supplementary Fig. 14 and Supplementary Fig. 17.

7. Comments: *“To prove the stability of device, authors needs to show long-term stability of GOx or Ex-4”.*

Response: Thanks for the suggestion. As m-GOx did not dissolve during treatment, we evaluated the long-term enzyme activity by detecting TMB oxidation (Fig. 3e, Supplementary Fig. 3). Moreover, the long-term stability of Ex4 was assessed by CD, HPLC and mass analysis (Fig. 4g, Fig. 4h and Supplementary Fig. 3). We have added the discussion in the text; please see highlights on Page 12, 14.

8. Comments: *“In addition, no change of chemical structure should be checked by Mass Spectroscopy instead to HPLC”.*

Response: Thanks for the comment. We have included mass analysis (Supplementary Fig. 8) for the released Ex4. Results indicated no degradation occurred and only a small proportion of thiol groups (<5%) was oxidized during the release process. We have added the discussion in the text; please see highlights on Page 14.

9. Comments: *“Authors need to include IACUC approval number for animal experiments”.*

Response: Thanks for the suggestion. The IACUC approval number is NIBIB 15-01. We have added it in the text; please see highlights on Page 32.

10. Comments: *“All quantitative experiments need repeat number and statistical analysis to prove the significant difference with method name”.*

Response: According to the suggestion, we have provided repeat number and statistical method for all the quantitative data analysis. Please see Fig. 3e, Fig. 3g, Fig. 4b, Fig. 4e, Fig. 4f, Fig. 6b, Fig. 6c, Fig. 6e, Fig. 6f, Fig. 6h, Fig. 6k, Fig. 6l, Supplementary Fig. 14 and Supplementary Fig. 17. We also provide the description in the text; please see the highlights on Page 33.

Minor revisions

1. Comments: *“Please change the reference No.2 from website to journals because website*

can be easily changed”.

Response: Thanks for the suggestion. We have changed the Ref #2 to a journal.

Reviewer #2

Major comments

1. Comments: *“The authors propose that a microneedle system could be used as a smart glucose responsive fashion, where the release of Ex4 would help maintain glycemic control. However, for clinical translation, it is unclear from the manuscript what therapeutic dose of Ex4 will be required and if this could be successfully loaded into a microneedle system i.e. with the loading efficiency in their system would they be able to achieve a therapeutically relevant dose in human and what size would the patch be? Are there any limitations of the animal model where the thickness of skin and proximity to blood glucose could impact on the efficacy of this system in clinical use? This information should be added to the discussion”*

Response: It is a very important comment. Currently, we only tried our system in a mouse model. There are significant differences in pharmacokinetics, pharmacodynamics, skin thickness between human and mice. Therefore it is our next-step plan to optimize the MN length, shape, Ex dose and administration frequency in primate model to achieve stable therapeutic blood Ex4 concentration. It should be mentioned that commercialized long-acting Ex4 formulation loaded with 0.8 or 2 mg Ex4 was administered once weekly in patients (Ref #59, 64). Our patch could load at least 0.3 mg Ex4, and is able to maintain 5-day normoglycemia and a slow BGL increase to the original level in 9 days. Although there are still some challenges for clinical translation, we believe our patch can serve as a promising candidate for T2D therapy in the future. We have added the discussion in the text; please see the highlights on Page 26.

2. Comments: *“Control of injection under skin? Without MN”*

Response: According to the comment, we have added a control group (m-Ex4/m-GOx without MN) in the animal study; please see Fig. 6b, Fig. 6c, Fig. 6g and Fig. 6h. We also evaluated the inflammatory reaction induced by MN-array patch treatment or intradermal injection (Supplementary Fig. 14). We have added the description in the text; please see the highlights on Page 19-20.

Other comments

1. Comments: *“The schematic of figure 1 is not immediately clear and could be enhanced”*

Response: Thanks for the suggestion. We have improved Fig. 1.

2. Comments: *“Supplementary Figure S10 – the figure legend should provide additional detail.”*

Response: Thanks for the suggestion. We have added more description in figure legend of Supplementary Fig. 10.

3. Comments: *“The authors mention FDA approval for alginate products. References for the most the most relevant products should be included.”*

Response: Thanks for the comment. We have included the reference to support our statements. Please see Ref #67.

Reviewers' comments:

Reviewer #1 (Remarks to the Author):

The manuscript was clearly improved in terms of originality, scientific meaning, and impact. Many concerns were resolved thank to the author's efforts. The remaining most important question is the possibility that this technique can be applied to human. For examples, repeated use of this technique might induce side effects (immunogenicity) or reduced therapeutic efficacy? Or, drug loading efficacy of a MN patch is sufficient for human dose? Please let us know about answers using big animal experiments or clinical trial results using similar particles or microneedles. This answers will clearly improve the impact of this manuscript. In addition, please revise manuscript following the Nature Communication format (abstract: about 150 words, main text: no more than 5, 000 words). And, please revise typos, for example, italic for *in vitro* and *in vivo*, and (6b,c) in the line 8 of page 20.

Reviewer #2 (Remarks to the Author):

I am satisfied that the reviewer comments have been addressed and this manuscript is now suitable for publication

Point-by-point Response

Reviewer #1:

1. Comments: *“The manuscript was clearly improved in terms of originality, scientific meaning, and impact. Many concerns were resolved thank to the author’s efforts. The remaining most important question is the possibility that this technique can be applied to human. For examples, repeated use of this technique might induce side effects (immunogenicity) or reduced therapeutic efficacy? Or, drug loading efficacy of a MN patch is sufficient for human dose? Please let us know about answers using big animal experiments or clinical trial results using similar particles or microneedles. This answers will clearly improve the impact of this manuscript. In addition, please revise manuscript following the Nature Communication format (abstract: about 150 words, main text: no more than 5, 000 words). And, please revise typos, for example, italic for in vitro and in vivo, and (6b,c) in the line 8 of page 20.”*

Response: Thanks for the comment. To address the potential to translate the MN-array patches to clinical applications, we added more discussions in the manuscript (please see the “Discussion” section). There has been literature report that repeated MN-array patch treatments did not alter skin appearance or barrier function, and caused no measurable disturbance of serum biomarkers of infection, inflammation or immunity, suggesting high biocompatibility of this microneedle platform. Please see Ref#61. Moreover, multi-time treatments by Ex4 formulations including macroparticles or MN-array patch have proven to be able to induce satisfactory therapeutic effect both in the mouse model and in patients. Please see Ref#54 and Ref#29. Therefore, we believe multi-time treatments by our MN-array patch will unlikely induce major side effects and reduce therapeutic efficacy. As for the dose, the most successful long-acting formulation for Ex4 delivery is BYDUREON®. Although now it contains 2 mg Ex4 for once-weekly administration, the 0.8 mg Ex4-loaded formulation also shows high therapeutic potentials. Please see Ref#59. Our dose (300 µg) is not much lower than the commercial formulations. Moreover, we are developing longer MNs (~1500 µm) to load a higher dose that we believe will exceed 1 mg Ex4 per patch. We plan to optimize the MN design and administration frequency for clinical translation of our MN-array patch-based strategy. We have included all the discussions in the manuscript. Please see highlights on pages 17-18. We also revised the manuscript

following format checklist for Nature Communications, and corrected some spelling mistakes.

Reviewer #2

1. Comments: *“I am satisfied that the reviewer comments have been addressed and this manuscript is now suitable for publication”*

Response: Thanks for the recommendation.